# Beyond Prescriptions Monitoring Programs: The Importance of Having the Conversation about Benzodiazepine Use

**DOI:** 10.3390/jcm8122143

**Published:** 2019-12-04

**Authors:** Erin Oldenhof, Jane Anderson-Wurf, Kate Hall, Petra K. Staiger

**Affiliations:** 1School of Psychology, Deakin University, Geelong, VIC 3220, Australiakate.hall@deakin.edu.au (K.H.); 2Reconnexion, Malvern East, Melbourne 3145, Australia; jane.andersonwurf@reconnexion.org.au; 3Centre for Drug Use, Addictive and Anti-social Behaviour Research (CEDAAR), Deakin University, Geelong, VIC 3220, Australia

**Keywords:** benzodiazepines (BDZs), long-term use, dependence, prescription monitoring programs (PMPs), deprescribing, overdose deaths

## Abstract

Internationally there is an escalation of prescription-related overdose deaths, particularly related to benzodiazepine use. As a result, many countries have implemented prescription monitoring programs (PMPs) to increase the regulation of benzodiazepine medications. PMPs centralize prescription data for prescribers and pharmacists and generate alerts to high-doses, risky combinations, or multiple prescribers with the aim to reduce inappropriate prescribing and subsequently the potential of patient harm. However, it has become clear that prescribers have been provided with minimal guidance and insufficient training to effectively integrate PMP information into their decision making around prescribing these medications. Accordingly, this paper discusses how PMPs have given rise to a range of unintended consequences in those who have been prescribed benzodiazepines (BDZs). Given that a gradual taper is generally required to mitigate withdrawal from BDZs, there are concerns that alerts from PMPs have resulted in BDZs being ceased abruptly, resulting in a range of unintended harms to patients. It is argued that best practice guidelines based upon a patient-centered framework of decision-making, need to be developed and implemented, in order to curtail the unintended consequences of PMPs. This paper outlines some key considerations when starting the conversation with patients about their BDZ use.

## 1. The Rise of Benzodiazepines

Nearly sixty years ago the first chemical compound (chlordiazepoxide) of the benzodiazepine (BDZ) drug class was discovered, heralding a new era of anti-anxiety medication. The earliest iterations, traded as Librium and Valium, seemed untroubled by the risks and unpleasant side effects of their predecessors, the barbiturates, and soared in popularity amongst prescribers becoming a panacea for everyday stress [1,2]. This ascendancy of Valium was apparent for over a decade, during which time it continued to be the most commonly prescribed psychotropic medication, and saw it become the first to ever reach billion-dollar status [3,4]. 

During this period of favor, however, strong evidence was arising of the adverse effects associated with BDZs, in particular the risk of dependence. Professor Malcolm Lader, one of the first to actively disseminate such findings, warned that the risk of dependence was evident even in those who were prescribed doses considered low and therapeutic [5]. Around that time another important figure, Professor Heather Ashton was collating reports from hundreds of individuals attempting to withdraw from BDZs, culminating in the seminal guidelines for BDZ withdrawal—The Ashton Manual [6]. The now established risk of dependence was documented in prescribing guidelines by 1988, recommending that BDZs be prescribed only in the short term, between 2 and 4 weeks [7]. Yet this had little impact on prescribing practice, with BDZs well-established in the prescriber’s repertoire and patients’ mindsets as the go-to medication to relieve anxiety and sleep issues. Long-term use was already widespread, and the risks associated with long-term use were not communicated to the consumer in the form of public health messages. 

## 2. Costs of Benzodiazepine Use

Since that time evidence of broad-ranging harms resulting from long-term BDZ use has continued to grow, including cognitive deficits [8,9,10], reductions in quality of life [11,12,13], depression [14,15,16], risk of suicide [17], risk for road accidents [18,19], and specific to older adults—the risk of falls [20,21,22], developing dementia [23,24,25], and of mortality [26,27,28]. Short-term, paradoxical effects such as increased anxiety, agitation, disinhibition, and aggression have also been observed [29,30]. Despite overwhelming evidence that over-time harms associated with BDZs may come to outweigh the benefits, the prescribing of BDZs long-term is endemic, and a problem reported globally [31,32,33,34,35,36]. In 2011, Professor Malcolm Lader asked the question “will we ever learn?”, expounding the systemic issue of prescribers and regulatory bodies consistent disregard of prescribing recommendations [37].

It was not until more alarming evidence emerged that BDZs were regularly involved in overdose deaths that system-wide responses were instigated. Since the early 2000s, Australian data have shown that prescribed medications rather than illicit substances were the leading contributors to overdose deaths [38,39], a trend also observed in North America and Europe [40,41,42,43,44]. The concerning nature of these findings has increasingly led to greater regulation through the implementation of prescription monitoring programs (PMPs), a government level response that this issue demanded. By centralizing prescribing and dispensing data, PMPs are an effort to create stewardship over inappropriate prescribing (such as multiple prescribers, high-doses, and risky drug combinations) in order to curtail rising mortality rates. Despite decades of repeated warnings, it has taken this top-down response to bring about greater duty of care over these drugs of dependence. 

## 3. Prescription Monitoring Programs: Impacts and Implications

PMPs have existed in the US for decades, and while successful at reducing the prescribed medications being monitored [45], have not convincingly addressed the underlying harms or problematic prescribing practices as intended [46,47,48,49]. Although primarily targeting prescription opioids, evidence indicates that when PMPs include BDZs they are also ineffective at reducing BDZ-related deaths [50,51]. As PMPs vary significantly state by state in the US, it is difficult to surmise why a reduction in prescriptions has not translated into diminished harms. In any case, one clear lesson from the States is that mandatory use of PMPs produces more consistent results, and appears to reduce multiple provider episodes or the incidences of “doctor shopping” [52,53].

In Australia, the first PMP targeting prescription opioids was introduced ten years ago in Tasmania, and with data available in real-time there were also promising findings around the reduction in overdose deaths and doctor shopping [54,55]. That said, despite repeated warnings of an impending “opioid crisis” [56], when heroin-induced deaths are parceled out from Australian mortality statistics, BDZs and not opioids are consistently the leading pharmaceutical implicated in overdose deaths [57]. Unsurprisingly, when the second state (Victoria) planned a PMP and investigated the harms caused by prescription medications, it concluded unequivocally that BDZs (and z-drugs) should be monitored [39]. And so at the beginning of 2020, Victoria will implement one of the first mandatory, real-time PMPs in the world that also monitors BDZs [57]. With this on the horizon, we must take time to understand why PMPs have not reduced patient harms to the extent anticipated. In addition, and more importantly, we must heed cautions that PMPs may generate new, reflexive harms [39]. These issues require careful consideration in order to avoid repeating preventable harms, particularly for those already involved in long-term use of BDZs.

### PMPs and Unintened Patient Harm

PMPs produce alerts to “at-risk” patients and offer details of their prescription history, however they do not give guidance on what to do with this new information, so it is argued that the prescriber’s response to these data is what gives rise to unintended consequences. In the literature, two themes emerge around how prescribers react to limitations imposed on their prescribing, and have been characterized as the “substitution effect” [58] and the “chilling effect” [59]. The substitution effect occurs when a prescriber responds to increased restrictions on monitored medications by compensating with the use of other, unmonitored medications. This was evidenced when the increased regulation of schedule II opioids in three US states, concurrently saw an increase in schedule III opioids being prescribed [60]. This highlights why an overall reduction in schedule II opioids did not result in diminished harms associated with opioids in general. In the case of BDZs there are limited alternatives available, so prescribers tend to compensate with off-label medications, such as sedating antipsychotics or z-drugs in place of a BDZ for sleep [61,62,63]. Consequently, when prescribers compensate in response to restrictions it can lead to suboptimal treatment and expose patients to greater risks and side effects. Another example of the substitution effect was seen in Australia when attempts to curb the quantity of Alprazolam being prescribed was actioned by removing large pack sizes and 2mg tablets from the pharmaceutical benefits scheme (PBS). This saw prescribers compensate with increased private prescriptions, prescribing greater pack quantities, and making more requests for authority to prescribe [64]. Taken together, this suggests that putting restrictions on prescribers without clear guidance on how to manage these limitations, may drive compensatory behaviors that contribute to the unintended harms of PMPs. 

The other observed response to PMPs, the “chilling effect”, occurs when patients who genuinely require their prescription, are ceased inappropriately [46,65,66,67]. Refusal to prescribe may occur when the patient is on a higher than recommended dose or on a risky combination of medications, that requires greater supervision and consequently greater onus on the prescriber. Paradoxically, studies in the US show that PMPs have a greater negative impact on those who used their BDZs appropriately than those who did not [67]. As a result, serious concerns have been expressed that PMPs may prevent patients from accessing medications they legitimately need, from the prescriber’s fear of being “flagged in the system” for inappropriate use [68,69]. Furthermore, when faced with limited access to their prescribed medication some patients may resort to illicit alternatives, which may in part explain the minimal impact on overdose deaths [70,71,72]. Patients’ desperate attempts to seek medication from alternate means only increases their risk, as unregulated sources could lead to greater harms.

Another example of the chilling effect occurs when prescribers cease a prescription based on the PMP revealing multiple prescriber involvement, without giving due consideration to other important factors. In fact, some prescribers appear to use PMPs to detect patient “dishonesty”, and with proof of multiple prescribers promptly “drop” their patients [73,74]. For patients prescribed BDZs, being abandoned by their prescriber puts them at risk of experiencing acute withdrawal, as coming off BDZs cold turkey can lead to severe withdrawal symptoms and increases the likelihood of protracted withdrawal [6,75,76,77,78]. This may also account for the increase in BDZ-related hospital presentations in the wake of PMPs [79]. A grave concern is that patients identified with multiple prescribers are often those with several chronic health conditions, and these vulnerable populations have historically been disproportionately affected by PMPs [80,81]. Prescribers’ response in these situations therefore needs a lens beyond that of a “doctor shopper”, as multiple prescriber involvement may simply reflect poor care-co-ordination [82], where patients see numerous prescribers for multiple and valid reasons. 

In Victoria, the overwhelming number of BDZ-related overdose deaths are from legally prescribed medications, and less than 25% of these involved multiple prescribers [82]. This highlights why PMPs as a stand-alone measure are inadequate to correct harms associated with prescription medications. It also sheds light on why increasing regulation can give rise to new issues, where patients are prescribed alternative and less than adequate medications, or worse, cut off completely. For decades’ prescribers have failed to actively manage the harms of BDZs, and additional pressure from PMPs highlights how under-prepared they are to address this long-standing issue. It is proposed that attempts to reduce the more malignant harms of BDZs, overlooks the issue of long-term use and means that more insidious harms, such as dependence, remain untreated. Current evidence indicates that some prescribers are side-stepping the issue of long-term use of BDZs either by compensating with other medications or refusing to continue prescribing. Neither approach directly addresses the issue, which highlights an obvious need for the conversation to begin between prescriber and patient regarding long-term BDZ use.

## 4. The Prescriber’s Dilemma

Despite widespread implementation, PMPs on their own are blunt instruments that poorly target the harms for which they were intended. As a result, PMPs place prescribers in a dilemma, on one hand forcing them to reduce the prescribing of BDZs, but on the other, offering little advice on how to respond to alerts of potentially risky medication use [51]. Prescribers are left to grapple with the long-standing issue of BDZ dependence on a case-by-case basis, and this environment has unmasked tensions between patient stigma and the prescribers’ duty of care [83]. More specifically, when required to manage a potentially inappropriate prescription, prescribers’ decision-making is often guided by the patient–prescriber relationship [84], meaning they may decide to cease a prescription if they doubt a patients’ intention, or if they had a prior negative experience with the patient. This is not surprising given the lack of guidance on managing competing needs of a patient dependent or “misusing” BDZs, and a prescriber held to account for dispensing a medication flagged as inappropriate by a PMP. With new information increasing patient complexity, prescribers are likely to face this dilemma more regularly and increase patient harm if left unaddressed. 

When patient care becomes contingent on prescriber perceptions, this increases the potential harms to those who are dependent or misusing their BDZ. Importantly, dependence on BDZs is clinically distinct from other forms of drug dependence, as it is often iatrogenic in nature, characterized by low-doses and without dose escalation [85,86,87]. This means that it can often go unnoticed and increases the likelihood that it is disregarded. The misuse of BDZs is also thought to be unique to this drug class, as those identified misusing BDZs do so infrequently and in a manner consistent with the prescribed indication (i.e., to cope) [88]. This is found to be true whether the misuse is in the context of higher-risk, polysubstance use or not. Accordingly, both dependence and misuse need to be de-stigmatized in the prescribers’ mind, to stop unhelpful perceptions misguiding responses that are harmful to patients. This population is also vulnerable to falling between the cracks of mental health and addiction services, as there is little in the way of specialist support for those experiencing BDZ dependence. Put simply, stigma associated with BDZ dependence and misuse needs to be tackled at the front-end, otherwise it will remain a barrier to a patient-centered approach in managing difficult decisions prompted by PMPs.

### 4.1. Steps towards Patient-Centred Care 

The need for additional guidance and training is evident, as prescribers need support around decision making in complex situations, and a need for effective responding is only compounded by the time-limited nature of their work. At its core, this must include a shift in the framing of the decision-making process towards judgment of the necessary treatment and not of the patient [89]. The population being flagged by PMPs for BDZ misuse will likely be highly heterogeneous, meaning the patient response will be varied with a range of factors requiring consideration to develop the most appropriate plan [90]. In some cases, it may be necessary to continue prescribing while undertaking a medication review, in collaboration with other treatment providers, which puts prescribers in a position of holding short-term risk. Such situations are challenging, and these competing demands are taxing for prescribers on both a professional and personal level [91]. To that end, a clearer understanding and documentation of best practice management is necessary to navigate decision making, and to reduce the impact of PMPs stigmatizing patients [92]. Prescribers wanting to support their patients are left unsupported by the system to manage this complexity and risk, therefore regulatory and peak bodies need to play a larger role in the implementation of PMPs. 

Prescribers are also confronted with the challenging issue of how to have the conversation about BDZ misuse and dependence. Research shows the manner in which information produced by PMPs is conveyed has significant impacts on patient health outcomes [73], and accordingly requires specific attention. Principles of patient-centered care are widely disseminated in health care, yet it is unclear how this ethos will be translated through PMPs. The very process of increasing a prescriber’s control and access to patient information is in marked contrast to core principles of shared decision-making and patient consultation [93]. Along with communication and interpersonal skills, patient empathy and autonomy need to be given highest priority alongside training for the implementation of PMPs [94]. These challenges to the patient–prescriber relationship parallel experiences in chronic pain management, where the paucity of research illustrating how to integrate patient-centered approaches with new guidelines and practices of PMPs are echoed [95,96]. 

### 4.2. Having the Conversation

Notwithstanding, PMPs do create an opportunity to initiate challenging conversations that could lead to greater awareness of risks associated with BDZs [74]. A series of targeted conversations prompted by PMPs could help reduce both the unintended harms created by PMPs as well as the specific harms they aim to reduce. We offer some guidance on how these conversations may flow over a period of time (see Figure 1) noting that one-off, brief conversations are insufficient to address long-term BDZ use. As a community we should aspire to support all patients and prescribers to create the opportunity to engage in more patient-centered conversations. It is envisaged that such conversations would seek to increase patients’ knowledge and awareness of harms associated with taking BDZs, and then encourage patient agency to share in the process of actively managing identified harms. This would denote significant progress in managing BDZ-related harm, as starting conversations to collaboratively review the cost-benefit of BDZ use can only enhance the effectiveness of PMPs in reducing overdose deaths. The BDZ-specific guidelines of The Royal Australian College of General Practitioners (RACGP) state that the therapeutic alliance between patient and prescriber is critical to the process of discontinuing BDZs, but again, offer little guidance on how to affect this relationship [97]. Subsumed more broadly by patient-centeredness, lessons from psychology tell us that the therapeutic alliance is enhanced by personal attributes such as warmth, trust, and understanding, along with skills of active listening and exploration of the patient experience [98,99]. Recent findings support the importance of these factors on the patient–prescriber relationship involved in long-term BDZ use, and that the quality of this relationship determined how willing a patient was to collaborate with and heed the advice of their prescriber [100]. Communication and interpersonal skills therefore require due consideration as key factors to enhancing the therapeutic alliance and espousing collaborative care.

Importantly, prescribers also need to identify reasons why an ongoing prescription for a BDZ may be indicated, to ensure PMPs do not impede the rational prescribing of BDZs [97,101]. In the case where discontinuation is deemed appropriate, it is important to incorporate the “stages of change” approach commonly employed in healthcare [102]. The process of stopping BDZs takes time and needs to be an ongoing conversation, not a one-off, in order to give a patient time to prepare for withdrawal. Thus, information on hand for the patient to read and return to discuss on a later appointment is preferred, as well as information about the reduction process which involves a gradual taper over time. Patients must be involved in the decision-making process when they start, as the rate of reduction will vary greatly and may require additional psychological support. Coordination of care is essential for many withdrawing off BDZs, and for those who have relied on BDZs long-term, adjunctive cognitive behavior therapy (CBT; or CBT-I) [103,104,105,106] is shown to enhance success rates and decrease the likelihood of relapse [106,107]. Victoria perhaps has the best opportunity to offer a stepped-care model, as at present it is the only state in Australia that has a BDZ-specialist service, and has shown efficacy in a shared-cared approach with prescribers, to include a gradual taper alongside CBT [108]. 

## 5. Future Directions

The long-term use of BDZs has been an issue unheeded for decades, and the increased regulation brought about by PMPs brings this dilemma to the forefront. A careful balance needs to be struck between patient needs, who in good faith were prescribed BDZs long-term, with the knowledge that these medications are potentially creating or exacerbating harms. It is essential that prescribers are given the support they need to ensure PMPs do not continue propagating harm. Moreover, to successfully transform how BDZ dependence and misuse is viewed and treated, it requires input from multidisciplinary stakeholders to produce a comprehensive and system-supported response. For example, capitalizing on PMPs sharing of information may help to foster more collaborative care between prescribers and pharmacists [74], offering another means to offset the unintended harms of PMPs. 

A recent study investigating an e-module embedded alongside prescribing software was shown to effectively change prescribing patterns of BDZs [109]. Whilst promising, this intervention only focused on reducing initial BDZ prescriptions, so at present it remains unclear how a similar intervention might support prescribers to address issues around inappropriate and long-term use of BDZs. Nonetheless, any prescriber-led intervention needs to be time efficient, therefore an e-module that is streamlined within the prescriber’s workflow represents one possible strategy. Such a program would need to include prompts to assess risk factors and provide patient education, that at the same time offers clear and brief guidance that can be promptly implemented. Other important components of an e-module would be language that de-stigmatizes BDZ dependence and misuse, and shares the responsibility surrounding the likelihood of dependence. Prescribers require this support to shift away from traditional patient indicators of adherence, compliance, and trustworthiness towards empathy, understanding, and trust [110]. Reducing stigma towards patients dependent on their BDZs may offset more punitive approaches to the use of PMP data, but also reduces the risk of further stigmatizing patients [83,111]. Prescribers also need to feel confident in referring to and collaborating with specialist services available to their patients [108], and not further stigmatize their patients through referrals to more general alcohol and drug services [112].

The most convincing evidence of successful outcomes from PMPs are within regions that recognized the need for organizational support, prescriber education, and automated decision-making programs [113]. Training for prescribers should be explored through a range of alternate mediums such as video vignettes, face to face workshops, and interactive webinars, and focus on modeling conversation styles to expand skills for navigating patient conversations, with opportunity to practice these skills with peers. Organizations need to develop and offer these training opportunities and incentivize their uptake through professional development programs. These can only enhance the confidence of prescribers and equip them with a communication skillset necessary for these challenging conversations. To move beyond the “at-risk” use of PMPs, where prescribers only use PMPs in response to alerts, the focus needs to shift towards starting conversations. In Victoria, many patients who are at risk will not be captured by the planned PMP, which only creates notifications in response to high doses of opioids, risky combinations, and patients seeing more than four prescribers within 90 days. If the conversation is not started, Victoria may also see a rise in unintended harms and minimal impacts on overdose deaths, like so many PMPs before.

## 6. Conclusions

The aim of PMPs is to address inappropriate prescribing practices and reduce overdose deaths, amongst a myriad of other negative consequences of BDZ use. However, in order to achieve this, the question of how prescribers should use the information provided to them needs careful consideration and planning, supported by appropriate guidelines and training. PMPs are a necessary impetus for change and may finally end the era of overlooking guidelines, but we argue that they do not work as a stand-alone intervention. Bringing about a cultural shift in attitudes to prescribing and to address issues such as stigma are obstacles that will not be overcome in the short-term, but nonetheless must be confronted. It is essential that broader principles in patient-centered care are embraced and adopted alongside PMPs in order to mitigate their unintended harms. Difficult conversations need to occur. To successfully navigate these conversations, prescribers need additional guidance alongside PMPs to manage challenging decisions in a way that is patient centered. 

## Figures and Tables

**Figure 1 jcm-08-02143-f001:**
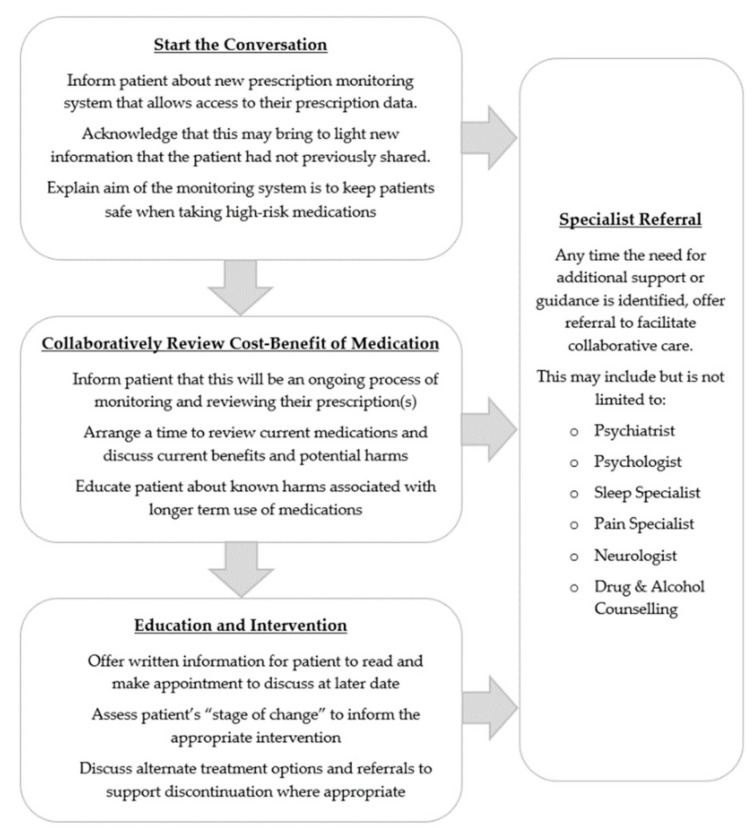
A patient-centered model of prescriber–patient conversations regarding long-term use of benzodiazepines (BDZs).

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
