# Peer review of "Beyond Prescriptions Monitoring Programs: The Importance of Having the Conversation about Benzodiazepine Use"

_jcm, 2019, doi:10.3390/jcm8122143_

Round 1
Reviewer 1 Report
The authors have done a significant job to provide some rigorous and critical thinking about the current situation of overdose death related to BDZs use on the context of implementation of PMPs. The authors has proposed constructive solutions to put forward the patient-centered framework of decision-making, prescriber education and conversation to mitigate the withdrawal of BDZs preventing unintended harms alongside with PMPs. I think this manuscript is of high-quality with innovative insight for practical issue-solving and research purpose.
Author Response
Response 1: We thank the reviewer for their very positive comments and the time taken to review this manuscript.
Reviewer 2 Report
Very good writing of the literature review regarding Prescription Monitoring Programs. There are some minor errors of syntax/grammar. See lines 83; 91; 118; 150;212.
The reader would recommend adding a brief discussion of current guidelines of rational drug prescribing. For example, see the APA Practice guidelines for treating people with Panic Disorder. South Australia Health offers guidelines when to prescribe benzos to balance the points presented throughout the manuscript. It may also be helpful to the reader's ease of reading to present a visual model of patient-provider relationship that can be included in section 4.2. It will break up the excess text.
The reader recommends updating reference item # 40. The NIH on drug abuse reports a ten-fold increase in benzo related Overdose death from 1999-2017.
Author Response
Response 1: We thank the reviewer for their time, and diligence in reviewing this manuscript, and we have addressed the grammatical errors that have been highlighted. We have also gone through the whole manuscript with a final proofread.
Response 2: We agree with this comment and have included a brief discussion pertaining to the rational use of BDZs (see page 5). It is important that the message the authors conveyed in this paper did not vilify the prescribing of BDZs, and accordingly thank the reviewer for this helpful suggestion.
Response 3: This is a helpful suggestion. We have spent some time considering this and developed what we think synthesizes the key point from section 4.2 into a visual map. Again, another very practical suggestion and we thank the reviewer.
Response 4: We have changed the reference to reflect this more recent citation, and thank the reviewer for noticing this detail.